# A Single-Dose of a Polyphenol-Rich Fucus Vesiculosus Extract is Insufficient to Blunt the Elevated Postprandial Blood Glucose Responses Exhibited by Healthy Adults in the Evening: A Randomised Crossover Trial

**DOI:** 10.3390/antiox8020049

**Published:** 2019-02-24

**Authors:** Margaret Murray, Aimee L. Dordevic, Lisa Ryan, Maxine P. Bonham

**Affiliations:** 1Department of Nutrition, Dietetics and Food, Monash University, Notting Hill VIC 3168, Australia; Margaret.murray@monash.edu (M.M.); Aimee.dordevic@monash.edu (A.L.D.); 2School of Chemistry, Monash University, Clayton, VIC 3800, Australia; 3School of Science and Computing, Galway-Mayo Institute of Technology, Galway H91 T8NW, Ireland; lisa.ryan@gmit.ie

**Keywords:** glucose, insulin, polyphenol, circadian rhythm, night, algae, hyperglycaemia, hyperinsulinaemia

## Abstract

When healthy adults consume carbohydrates at night, postprandial blood glucose responses are elevated and prolonged compared to daytime.Extended postprandial hyperglycaemia is a risk factor for type 2 diabetes. Polyphenols are bioactive secondary metabolites of plants and algae with potential to moderate postprandial glycaemia. This study investigated whether a polyphenol-rich alga (*Fucus vesiculosus*) extract moderated postprandial glycaemia in the evening in healthy adults. In a double blind, placebo-controlled, randomised three-way crossover trial, 18 participants consumed a polyphenol-rich extract, a cellulose placebo and rice flour placebo (7:15 p.m.) prior to 50 g available carbohydrate from bread (7:45 p.m.), followed by three hours of blood sampling to assess glucose and insulin. A subset of participants (*n* = 8) completed the same protocol once in the morning with only the cellulose placebo (7:15 a.m.). No effect of the polyphenol-rich extract was observed on postprandial glycaemia in the evening, compared with placebos, in the group as a whole. However, in females only, peak blood glucose concentration was reduced following the polyphenol-rich extract. In the subset analysis, as expected, participants exhibited elevated postprandial blood glucose in the evening compared with the morning following the cellulose placebo. This was the first study to investigate whether a polyphenol intervention moderated evening postprandial hyperglycaemia. The lowering effect observed in females suggests that this warrants further investigation.

## 1. Introduction

Postprandial hyperglycaemia is characteristic of type 2 diabetes mellitus (T2DM), however it is often present prior to clinical diagnosis and this is known as impaired glucose tolerance (IGT) or pre-diabetes [1,2]. IGT is defined as a 2-h postprandial glucose concentration between 7.8 and 11 mmol/L [3]. The use of bioactive phytochemicals from plants and plant-based foods to moderate hyperglycaemia is a growing area of nutrition research [4,5,6]. Marine algae contain bioactive polyphenolic molecules with capacity to moderate postprandial hyperglycaemia, including phlorotannins, which are unique to marine algae [4]. Potential mechanisms of action have previously been reviewed [4] and include the inhibition of carbohydrate digestive enzymes, α-amylase and α-glucosidase, as demonstrated in chemical assay [4] and the alteration of hepatic enzyme (inhibiting glucose-6-phosphatase and phosphoenolpyruvate carboxykinase) to promote glycogen production and the removal of glucose from the blood, as demonstrated in a diabetic mouse model [7]. Marine algal polyphenols (MAPs) have also been shown to upregulate phosphorylation of AMPK (adenosine monophosphate-activated protein kinase), ACC (acetyl-CoA carboxylase) and Akt (a serine/threonine protein kinase) in diabetic mouse and rat models to increase the number of GLUT4 (glucose transporter 4) transporters at the cell membrane and increase glucose uptake at a cellular level [8,9].

Human trials have shown that polyphenols (from plants and algae) have potential to lower postprandial blood glucose [10,11,12,13,14], with one initial human trial showing reductions in postprandial blood glucose following treatment with MAPs specifically [15]. However, the timing of polyphenol intake and the resulting effects on postprandial glycaemia are fundamentally unexplored in the literature. Typically, acute postprandial studies have been conducted in the morning only, whereas there is increasing evidence to indicate a time of day effect of carbohydrate consumption on postprandial glycaemia.

A small number of tightly controlled postprandial studies in human participants have shown significantly elevated glycaemia at night compared to during the day [16,17]. This phenomenon is due to the influence of circadian rhythms on glucose metabolism. Glucose homeostasis is regulated in a diurnal rhythm based on the external light-dark cycle [18,19,20] and internal feeding-fasting cycles [19,20]. Typically, feeding occurs during the light phase of the cycle and fasting occurs during the dark phase of the cycle [19,20]. When situations arise that challenge the status quo, such as eating during times intended for fasting, this results in a misalignment between the light-dark cycle and the feed-fast cycle resulting in postprandial hyperglycaemia. Consistent and prolonged evening hyperglycaemia may be a key contributor to the risk of T2DM and cardiovascular diseases (CVD), particularly as observed in shift working populations, who often have no choice but to eat late into the night [20,21,22]. Strategies to modify evening postprandial hyperglycaemia may help lower the risk of T2DM and CVD [23].

There is also evidence to suggest a role for the timing of treatment in influencing health outcomes. For example, taking blood pressure medications in the evening, rather than the morning, reduces overall risk of cardiovascular events in people with T2DM, largely through reducing blood pressure while asleep [24]. Similarly, using polyphenols to moderate postprandial hyperglycaemia at night [18,20,21,25], may help to improve blood glucose regulation and reduce the risk of T2DM and CVD.

This study investigated whether a polyphenol-rich extract from the marine macroalga *Fucus vesiculosus* moderated postprandial blood glucose and plasma insulin responses in healthy adults in the evening. It was hypothesised that healthy adults would exhibit postprandial hyperglycaemia in the evening, compared with the morning, and that the polyphenol-rich extract would reduce postprandial blood glucose, compared with placebo. A secondary outcome was the investigation of the influence of ethnicity and sex on postprandial responses.

## 2. Materials and Methods

### 2.1. Trial Design

A double-blind, placebo-controlled, randomised crossover trial was carried out in Melbourne, Australia from February 2017 to April 2018. This trial was registered with the ANZCTR, registration number ACTRN12616000126415p and is reported according to the CONSORT 2010 checklist (Appendix A). Ethical approval was obtained from Monash University Human Research Ethics Committee, approval number CF16/53–2016000019. All procedures were carried out in accordance with the Declaration of Helsinki, with written informed consent given by all participants.

### 2.2. Participants

Participants were recruited from the public via online advertising and fliers. Volunteers were normotensive males and females, aged 18–65 years, with fasting blood glucose (FBG) < 5.5 mmol/L and body mass index (BMI) between 18.5 and 30 kg/m^2^. Participants were excluded if they had been diagnosed with any gastrointestinal, liver or thyroid conditions, were taking medication for blood sugar control or hypertension, were taking natural health products known to have similar actions to polyphenols for example, fish oil, had undergone recent major surgery, were pregnant, planning a pregnancy or breastfeeding, consumed > 9 standard drinks per week or > 4 standard drinks per day of alcohol, were a smoker or had a cardiac defibrillator. Testing sessions were avoided during the week prior to the beginning of menstruation for all female participants.

In the absence of similar studies with a polyphenol treatment in the evening, power analyses were based on data from a meal timing study [17], which investigated postprandial blood glucose in the morning (8:00 a.m.), in the evening (8:00 p.m.) and at night (midnight), following an oral glucose tolerance test (OGTT). At 80% power, 15 participants were required to detect a difference in blood glucose incremental area under the curve (iAUC) of 150 mmol/L·2 h (G*Power 3.1.9.2 [26]). The recruitment target was 19 participants, to allow for up to 20% dropout.

### 2.3. Test Products

The intervention product was a 2000 mg dose of a powdered extract from the brown seaweed *F. vesiculosus*, containing 560 mg polyphenols and 1340 mg fucoidan (a complex carbohydrate) (Marinova Pty Ltd., Tasmania, Australia). Two placebo products were used, one was a 2000 mg dose of a cellulose fibre—Medisca^®^ Cellulose NF (Microcrystalline) (MEDISCA Australia Pty Ltd., New South Wales, Australia)—used to account for the fibre content of the seaweed extract. The other was a 2000 mg dose of commercially available rice flour (Ward McKenzie Pty Ltd., Altona, Victoria, Australia), which acted as the no treatment placebo. All test products were encapsulated in identical opaque size 0 capsules (The Melbourne Food Ingredient Depot, Brunswick, Australia).

### 2.4. Randomisation and Blinding

Computer generated randomisation was used to determine the order in which participants received the test products. Each supplement was coded with a corresponding letter to conceal its identity. The participants and the investigator carrying out participant enrolment, data collection and analysis (M.M.), were blinded as to which supplements participants received on each crossover. An investigator not involved in data collection (M.B.) carried out supplement order generation and allocation.

### 2.5. Procedure

Initial screening was conducted via telephone interview. Eligible individuals were then screened in-person at the research facility to determine BMI, blood pressure and fasting blood glucose concentration. Those who remained eligible were invited to join the study and allocated a supplement order. Participants attended three testing sessions, during which they received the polyphenol-rich seaweed extract, cellulose and rice flour supplements in a randomised, crossover manner. Participants received a pre-prepared meal (pasta with tomato-based sauce; 3425 kJ, 122 g carbohydrate, 3.3 g fat, 20.4 g protein) to consume between 8–9 a.m. on the morning of each testing day. They were asked to then fast until testing was complete, with the exclusion of water. Participants were also given a list of foods (naturally high in polyphenols) to avoid and asked to avoid strenuous exercise for 24 h prior to each testing session.

Participants arrived at the testing facility at 7 p.m. after a fast of ≥10 h. Two initial finger prick tests were taken to assess fasting blood glucose (at −45 and −35 min) and plasma insulin (−45 min). At −30 min (7:15 p.m.) participants were given a 2000 mg dose of either the polyphenol-rich extract, cellulose or rice flour. At 0 min (7:45 p.m.) participants were served 50 g available carbohydrate in the form of white bread (108 g) and asked to consume the entire portion within 7 min. Finger prick blood samples were taken at regular intervals to measure blood glucose (15, 30, 45, 60, 90, 120, 150, 180 min) and plasma insulin (30, 60, 90, 120, 150, 180 min) (Figure 1). This process was repeated three times in a crossover manner so that all participants received all three supplements. A one week wash out period was observed between each treatment.

A subset of eight participants also completed the testing protocol starting at 7 a.m. instead of 7 p.m. (with the standardised meal for dinner the night before) to confirm that elevated glycaemic responses were observed in the evening compared with the morning. In the morning, participants were only given the cellulose fibre placebo, as a comparator to the cellulose fibre placebo in the evening. Blood samples were collected for two hours instead of three because blood glucose returns to baseline levels within this time [17].

### 2.6. Outcome Measures

#### 2.6.1. Blood Glucose and Plasma Insulin

Capillary blood samples were obtained by pricking the fingertip with a Unistik^®^ 3 Extra single-use lancing device (Owen Mumford Ltd., Oxfordshire, United Kingdom). Three droplets of blood were wiped away prior to collection in a HemoCue^®^ Glucose 201 RT micro cuvette (Radiometer Pacific Pty Ltd., Mount Waverley, Victoria, Australia). Blood glucose concentrations were assessed immediately on collection of capillary blood samples using the HemoCue Glucose 201 RT System (Radiometer Pacific Pty Ltd., Mount Waverley, Victoria, Australia), according to standard procedures.

Plasma insulin was measured from capillary blood samples, which were collected using Safe-T-Fill^TM^ Capillary Blood Collection GK Systems containing EDTA anti-coagulant (item no. 077001, Kabe Labortechnik GmbH, Cologne, Germany) (pictured in Figure 1). At least 200 µL of whole blood was collected from the capillary at each time point. Whole blood was centrifuged at 4 °C and 1300 *g* for 15 min (serial no. 5703BI110739, Eppendorf AG, Hamburg, Germany). Aliquots of plasma was stored at −80 °C until analysis. Insulin concentration was measured using the Millipore ELISA Kits for Human Insulin (Cat. # EZHI-14K and EZHI-14BK, Merck Millipore, Bayswater, Victoria, Australia), according to kit instructions. Each sample was assessed in duplicate and absorbance measured using the Rayto Microplate reader (450 nm wavelength, RT-2100C, Abacus ALS, Meadowbrook, Queensland, Australia). The lowest detectable insulin concentration for this assay was 1.0 µU/mL, therefore any values below this were rounded up to 1.0 µU/mL. The highest accurately detectable insulin concentration was 200 µU/mL. Samples that read above this value were diluted 2:1, using assay buffer as a diluent, and re-run. Units were converted from µU/mL to pmol/L prior to statistical analysis. Across all plates, the mean coefficient of variation was 8.5% (standard deviation (SD) 15.3).

#### 2.6.2. Anthropometric Data

Height, weight and body composition were measured at the screening session. Participants removed shoes and socks for all anthropometric measures. Height was measured using the Harpenden Stadiometer (Holtain Ltd., Crymych, UK). Weight and body composition (% fat mass, % fat free mass, visceral fat (L)) were measured using the SECA mBCA 515 medical body composition analyser (SECA, Hamburg, Germany), with participants in light clothing. Waist circumference was measured over light clothing or bare skin at the narrowest point around the torso.

#### 2.6.3. Intolerance Symptoms

An intolerance symptoms questionnaire [27] was completed by participants 24 h after ingestion of each supplement to assess the occurrence and intensity of any side effects. Participants were asked to indicate whether they experienced intolerance symptoms as a result of taking the supplement (above any usual ailments) and whether they were of mild, moderate or severe intensity (scored as 1, 2 or 3, respectively). Side effects listed in the questionnaire were headache, anxiety, tiredness/exhaustion, lack of energy, tendency to become rapidly exhausted, reduction in appetite, increase in appetite, hiccups, nausea, vomiting, indigestion, stomach or abdominal pain, constipation, diarrhoea, gas, abdominal bloating, cardiac palpitations, balance disorders, reduced capacity to concentrate, feeling cold, muscle or joint pain, numbness, burning or itching sensations, dark or depressing thoughts.

#### 2.6.4. Diaries and Questionnaires

Food intake data were collected using a 3-day food diary (over two weekdays and one weekend day) which was cross-checked with participants and assessed using FoodWorks 8 (Xyris Software (Australia) Pty Ltd., Spring Hill, Queensland, Australia) to establish participants’ usual dietary intake. A food frequency questionnaire (FFQ) was used to estimate participants’ dietary polyphenol intake. The FFQ was adapted for the Australian diet from a British FFQ used to assess dietary polyphenol intake [28]. Daily dietary polyphenol intake was calculated using data from Phenol-Explorer 3.6 Database on polyphenol content in foods [29] and the United States Department of Agriculture (USDA) Database for the Flavonoid Content of Selected Foods, Release 3.1 (December 2013) [30]. The International Physical Activity Questionnaire (IPAQ) short version was used to assess physical activity habits. This questionnaire consists of four questions that assess the amount and intensity of physical activity and sitting time in participants’ daily lives [31]. These data contributed to the description of characteristics of the sample population.

### 2.7. Quantification of Soluble Polyphenols

An adapted Folin-Ciocalteu methodology was used to quantify the total soluble polyphenols in the extract [32] with phloroglucinol dihydrate used as standard (Sigma-Aldrich P38005, St. Louis, MO, USA). The extract was dissolved in 10 mL of distilled water and diluted to reach concentrations of 25, 50 and 100 µg/mL. The assay was performed by pipetting 2 mL of distilled water (blank), the phloroglucinol standards (5, 10, 15, 20, 30, 50 and 100 µg/mL) and the sample solutions into sequential vials. Folin-Ciocalteu reagent (500 µL) (Sigma-Aldrich F9252, St. Louis, MO, USA) was then added to each vial and allowed to stand for 5 min. Then 1500 µL of 7.5% *w*/*w* sodium carbonate solution and 4000 µL of distilled water were added to each vial. The reaction was then incubated in the dark at room temperature for two hours. Analysis was conducted using a spectrophotometer at 765 nm, with the solutions in quartz cuvettes. All samples, standards and blanks were run in triplicate and absorbance values were recorded.

### 2.8. Statistical Analysis

Analyses were performed using Statistical Package for Social Sciences (SPSS) version 22.0 (SPSS Inc., Chicago, IL, USA), with the level of significance accepted as *p* < 0.05. All results were assessed for normality using the Shapiro-Wilk test. Where results were not normally distributed the data were transformed, using the natural log and parametric tests applied. Where data could not be normalized, non-parametric tests were applied and data were reported as median (interquartile range (IQR)). Normally distributed data were reported as mean (SD). Postprandial blood glucose and insulin were assessed using iAUC and peak blood concentration. The iAUC was calculated using the trapezoidal method with baseline value removed and is expressed as mmol/L·3 h and pmol/L·3 h, respectively. Late phase insulin response was used as another measure of postprandial insulin and was calculated using the insulin iAUC from 30–120 min postprandial and is reported as pmol/L·90 min [17,33].

A one-way repeated measures analysis of variance (ANOVA) or Friedman’s Two-Way Analysis of Variance by Ranks test (for data that did not meet relevant parametric assumptions) were used to determine differences between the treatments for fasting glucose and insulin, iAUC, late phase insulin and peak concentrations. Supplement sequence, age, sex and % fat mass were added to the one-way repeated measures ANOVA as covariates when assessing iAUC and peak concentration. Differences between the treatments for intolerance symptoms were assessed using the Friedman’s test because the data was not normally distributed. For comparison between the population sub groups (female/male and Asian/Caucasian) independent *t* tests or independent samples Mann-Whitney U tests (for data that did not meet relevant parametric assumptions) were used to assess differences in fasting, iAUC and peak postprandial glucose and insulin concentrations. For the comparison between morning and evening, glucose and insulin were assessed by iAUC and peak blood concentration. Paired samples *t* tests or Wilcoxon signed-rank tests, were used to determine differences between morning and evening for iAUC and peak concentration. The morning evening comparison (iAUC) was from 0–120 min.

## 3. Results

### 3.1. Participants

Twenty-three participants were randomised, of whom 18 (12 women and 6 men) completed all three crossovers. Four participants withdrew prior to receiving any intervention and one participant withdrew after commencing the trial due to being unable to commit the time required for participation (Figure 2). Participants were 18–55 years of age with BMI ranging from 19.7–29.4 kg/m^2^. At the screening session, all participants had an FBG level below 5.5 mmol/L (Table 1). Five (28%) participants were of an Asian background (4 female, 1 male) and 13 (72%) of a Caucasian background (8 female, 5 male). Asian participants included those who self-identified as Chinese (*n* = 3), Indian (*n* = 1) or Indonesian (*n* = 1). Caucasian participants included those who self-identified as white Australian (*n* = 9) or British (*n* = 4).

Eight participants, a subset of the aforementioned 18 participants, completed both the evening and morning protocols and their characteristics are presented in Table 2. These participants were 20–55 years of age with BMI ranging from 19.9–27.7 kg/m^2^.

### 3.2. Comparison between Morning and Evening

Blood glucose iAUC was 2.2-fold higher in the evening (296.5 (104.8) mmol/L·2 h) than in the morning (132.8 (53.8) mmol/L·2 h) (*p* = 0.001). Peak blood glucose concentration was 0.3-fold higher in the evening (8.8 (1.6) mmol/L) than the morning (6.8 (0.7) mmol/L) (*p* = 0.009) (Figure 3). There were no differences in FBG levels (*p* = 0.569).

Plasma insulin results could not be normalised and therefore were reported as median (IQR) and analysed using non-parametric tests. Fasting insulin was higher in the morning than the evening (*p* = 0.02). There were no differences in postprandial plasma insulin iAUC between morning (14,877 (4887) pmol/L·2 h) and evening (16,349 (8619) pmol/L·2 h) (*p* = 0.263) nor for peak plasma insulin concentration between the morning (260 (127) pmol/L) and evening (258 (202) pmol/L) (*p* = 0.779) (Figure 3).

### 3.3. Effect of the Polyphenol-Rich Extract on Evening Postprandial Blood Glucose

There were no differences in FBG levels prior to the treatments (*p* = 0.077) (Table 3). There were no differences in postprandial blood glucose iAUC (*p* = 0.881) or peak concentration (*p* = 0.459) between the treatment arms (Figure 4). Supplement sequence (*p* = 0.623, 0.449), sex (*p* = 0.281, 0.162) and % fat mass (*p* = 0.804, 0.726) did not influence the effect of supplement on postprandial blood glucose iAUC or peak concentration. Participant age was a significant predictor of iAUC (*p* = 0.031), where those who were older had a higher iAUC, however it was not a predictor of peak concentration (*p* = 0.761).

There were no differences between those of Asian and Caucasian backgrounds for postprandial blood glucose iAUC (*p* = 0.118) or peak blood glucose concentration (*p* = 0.177) (Table 3). Asian participants had a higher FBG than Caucasian participants (*p* = 0.007), however all participants had a FBG level within the healthy range (< 5.5 mmol/L).

There were no differences between male and female participants for postprandial blood glucose iAUC (*p* = 0.506). Male participants exhibited higher peak blood glucose concentrations than female participants (*p* = 0.040), with significant differences observed following the cellulose (*p* = 0.024) and polyphenol-rich extract (*p* = 0.015) but not the rice flour (Table 3). Among female participants only, peak postprandial blood glucose concentration was significantly lower following the polyphenol-rich extract compared with the rice flour and cellulose (*p* = 0.018).

### 3.4. Effect of the Polyphenol-Rich Extract on Evening Postprandial Plasma Insulin

Plasma insulin results could not be normalised and therefore were analysed using non-parametric tests and reported as median (IQR). Prior to each treatment there were no differences in fasting plasma insulin levels (*p* = 0.978) (Table 4). There were no differences in postprandial plasma insulin iAUC (*p* = 0.801) or peak concentration (*p* = 0.411) between the treatment arms (Figure 5).

Across all three treatment arms, participants of an Asian background had significantly higher postprandial plasma insulin iAUC (*p* = 0.003 rice flour; *p* = 0.019 cellulose; *p* = 0.007 polyphenol), late phase insulin iAUC (*p* = 0.007 rice flour; *p* = 0.035 cellulose; *p* = 0.019 polyphenol) and peak plasma insulin concentrations (*p* = 0.002 rice flour; *p* = 0.046 cellulose; *p* = 0.007 polyphenol), compared with participants of a non-Asian background (Table 4). There were no differences between male and female participants for postprandial plasma insulin iAUC or for peak plasma insulin concentration (Table 4).

### 3.5. Intolerance Symptoms

The intolerance symptoms reported most frequently were tiredness/exhaustion, lack of energy and abdominal bloating. The tiredness and lack of energy were likely due to participants fasting throughout the day prior to the testing sessions. These symptoms were reported by participants following all three treatment arms. The majority of symptoms were reported to be of mild intensity (1 on an arbitrary scale of 0–3). There were no differences between the groups for any of the symptoms assessed, therefore any symptoms experienced were not likely to be a result of the supplement.

### 3.6. Polyphenol Content of Extract

The total soluble polyphenol concentration of the extract was determined to be 29.7%, according to the Folin-Ciocalteu assessment of water extracts of powdered *Fucus vesiculosus*. This concentration is comparable to the suggested 28% polyphenol content in the extract, as certified by the supplier. The mean coefficient of variation among the triplicates was 2.5%.

## 4. Discussion

Findings from this study support existing evidence that consuming carbohydrates in the evening causes an elevated glycaemic response, compared with the morning, due to the effects of circadian rhythms on glucose metabolism [3,20,21,25]. Whilst there was no overall lowering effect of the polyphenol-rich extract on postprandial glucose or insulin, the supplement was well tolerated in an acute setting and a significant reduction in peak postprandial blood glucose concentration was observed following the polyphenol-rich extract, compared with cellulose and rice flour, in females but not in males. The present study also identified significantly elevated postprandial plasma insulin responses in the evening among participants of an Asian background compared with a Caucasian background.

### 4.1. A Reduction in Peak Postprandial Blood Glucose Concentration Was Observed in Females but Not Males

This novel study identified that MAPs have potential to moderate glycaemic responses in the evening in females through lowering peak postprandial glucose concentration, an effect not seen among males. While this is the first known observation of sex differences in the anti-hyperglycaemic effects of polyphenols, this is consistent with literature indicating potential sex differences in the health-related effects of polyphenols and sex differences in glucose metabolism [34,35,36]. Females and males have significant physiological and hormonal differences, which have implications for the way in which the body maintains health, develops disease and responds to treatment [37,38,39,40], therefore it is important to investigate sex differences in health-related intervention trials. Differences between the sexes have not previously been examined in trials investigating the postprandial glycaemic lowering effects of MAPs [15,27,41].

In the present study, peak blood glucose concentration was reduced by 6% (from 8.4 mmol/L to 7.9 mmol/L) following the polyphenol-rich seaweed extract, compared with cellulose and rice flour, among females. The European Diabetes Policy Group targets for postprandial blood glucose control indicate that, to reduce arterial risk, postprandial peaks should not exceed 7.5 mmol/L [42]. While mean peak postprandial concentrations remained above this in the present study, this target highlights the potential value of reducing peak glucose concentrations. MAPs, including those from *F. vesiculosus,* may reduce the immediate spike in blood glucose concentration following carbohydrate consumption through the inhibition of α-amylase and α-glucosidase [4,43] and delaying the digestion of carbohydrates and absorption of glucose into the blood [44,45].

Another sex difference observed in the present study was that male participants had significantly higher peak postprandial blood glucose levels compared with female participants. This is in contrast to evidence which suggests that females are more likely to experience elevated blood glucose concentrations following an OGTT [40,46,47], as a consequence of generally having a smaller body size and thus less muscle mass but being given the same glucose load [47]. Likewise, in the present study, females had a smaller body size than males (on average 13.7 kg less fat free mass than males), which therefore does not explain the differences observed. It is also unlikely to be a consequence of age or visceral fat as there were no significant differences between males and females for age, BMI or visceral fat volume. This finding may be due to differences between the sexes in the way that glucose homeostasis is maintained [40]. Males typically have lower insulin sensitivity than females, which is countered by their greater muscle mass and lower body fat mass [40]. Reduced insulin sensitivity results in less efficient uptake of glucose by the tissues and less efficient inhibition of endogenous glucose production which may explain the elevated postprandial blood glucose responses observed in males [48].

### 4.2. The Polyphenol-Rich Extract Had No Effect on Evening Postprandial Glycaemia Overall

The elevated postprandial blood glucose responses observed here in the evening are consistent with existing literature and the effects of circadian rhythms on glucose metabolism [17,25]. However, the hypothesis that the polyphenol-rich *F. vesiculosus* extract would lower evening glycaemic response was not confirmed. Reductions in evening postprandial glycaemic responses have previously been observed following treatment with acarbose, an α-glucosidase inhibitor drug [49]. In healthy men, a 50 mg dose of acarbose successfully reduced postprandial glycaemic responses to a meal consumed at 7:00 p.m., compared with placebo, by inhibition of α-glucosidase [49]. In contrast, studies that compared blood glucose responses to a meal, consumed in the evening compared with the morning, observed that (1) consuming a low glycaemic index (GI) meal, compared with a high GI meal, resulted in equally elevated glycaemic responses in the evening [25] and (2) that postprandial glycaemic responses to a low GI meal are elevated in the evening, compared with the morning [17]. Consuming low GI meals is an established method for moderating glycaemic responses during the day. This highlights the challenge of overcoming the diurnal changes in glucose homeostasis, including the relative insulin insensitivity and glucose intolerance observed in the evening and suggests that delaying the breakdown of carbohydrates and absorption of glucose into the blood (the effect of low GI foods and one of the proposed actions of MAPs), may be ineffective to do so.

In the present study, participants fasted for 10 hours throughout the day prior to postprandial testing, during which time their blood glucose levels would have been maintained by endogenous glucose production. When participants were then fed carbohydrates in the evening, the prevailing blood glucose response was likely a combination of glucose from the meal as well as endogenous glucose [50]. While polyphenols have been shown to reduce endogenous glucose production and promote glycogen storage [7], there may be merit in examining the effects of a polyphenol-rich extract on continuously monitored glycaemic responses where meals are consumed at breakfast, lunch and dinner, to reflect a more real-life scenario and prevent unusually elevated endogenous glucose production caused by fasting.

Polyphenols from different algal species can have differing effects on postprandial glycaemic responses [14,51]. Studies that have used polyphenols from algal sources, such as *Ecklonia cava* and land-based sources, such as blackcurrants, have shown effective reduction of postprandial glycaemic responses in humans, when tested in the morning [10,11,15]. It may be that the algal extract tested in the present study and the particular polyphenols it contains are less effective at reducing postprandial glycaemic responses than polyphenols from other algal or plant species.

### 4.3. Asian Participants Experienced Exaggerated Postprandial Plasma Insulin Responses

An elevated postprandial insulin response among Asian, compared with non-Asian, individuals, despite no differences in blood glucose responses (iAUC or peak concentration), has been previously demonstrated in the morning [27]. Similar findings were observed in the present study, with no differences between ethnic groups for blood glucose responses (iAUC or peak concentration) but participants of an Asian background exhibiting markedly elevated plasma insulin responses in the evening, compared with Caucasian participants. Results from this small study support growing evidence that individuals of an Asian background are at an increased risk of reduced insulin sensitivity, with the potential to lead to an increased risk of T2DM [52,53,54].

Furthermore, in the present study, Asian participants had a mean 2-hour postprandial glucose concentration of 7.8 mmol/L or above following all treatments, whereas Caucasian participants had a mean 2-h glucose concentration of 6.7 mmol/L or lower following all treatments. The 2014–2015 General Practice Management guidelines for T2DM indicate that a 2-h postprandial glucose concentration between 7.8 and 11 mmol/L indicates a diagnosis of IGT, which implies reduced insulin sensitivity [3]. While it is well understood that insulin sensitivity decreases throughout the day and into the evening [25,55,56], these findings suggest that this may be exaggerated among Asian individuals, resulting in highly elevated postprandial insulin responses and 2-h glucose concentrations within the range for IGT in the evening among healthy individuals.

### 4.4. Strengths and Limitations

This was the first study to investigate the efficacy of a polyphenol-based intervention at moderating the postprandial hyperglycaemia that occurs in the evening. A strength of this trial was the randomised, double-blind, placebo-controlled, crossover nature of the study design. Furthermore, the use of both a cellulose fibre and non-fibre (rice flour) placebo accounted for the fibre (fucoidan) content of the extract to show the effect of the polyphenol content of the extract alone and provided a comparator to assess the effect of the extract as a whole. Another strength of this research was the investigation of postprandial responses based on ethnicity and sex. These findings indicated clear differences in insulin metabolism between individuals of Asian and Caucasian backgrounds and differences in peak postprandial glucose concentrations between males and females, even in a relatively small sample. In addition, the comparison between postprandial responses in the evening and the morning supports and builds on evidence for the effects of meal timing on metabolic outcomes [17,18,21,25]. A potential limitation of the study was the relatively small sample size (18 participants) which puts a limit on the applicability of the observations regarding ethnicity and sex until research with larger samples confirms these findings. Another potential limitation was the use of healthy participants, which may have diminished the glycaemic lowering potential of the polyphenol-rich extract due to individuals already having adequate glucose regulation. Further research in populations with pre-diabetes or T2DM may observe effects of greater magnitude. This research provides pilot data that future studies investigating polyphenol-based interventions on evening glycaemic responses can use to calculate power.

## 5. Conclusions

In line with the literature, an elevated postprandial blood glucose response was observed in healthy individuals when carbohydrates were consumed in the evening compared with the morning. Though not clinically significant, a lowering effect of the polyphenol-rich extract on peak postprandial glucose concentration was observed among females in the evening, compared with cellulose and rice flour, with no discernible intolerance symptoms. Further research should investigate the glycaemic lowering effects of polyphenols in the evening. If polyphenol treatment can moderate postprandial hyperglycaemia in a state of IGT in the evening, it may also help people diagnosed with IGT to manage their blood glucose levels and prevent progression to T2DM. This study identified that Asian participants exhibited elevated postprandial insulin responses in the evening, compared with Caucasian participants, highlighting the need for further research investigating the effect of ethnicity and meal timing on postprandial glycaemic and insulinaemic responses.

## Figures and Tables

**Figure 1 antioxidants-08-00049-f001:**
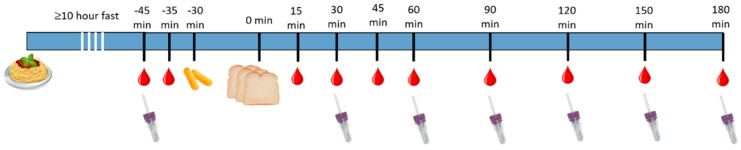
Procedure during each study session.

**Figure 2 antioxidants-08-00049-f002:**
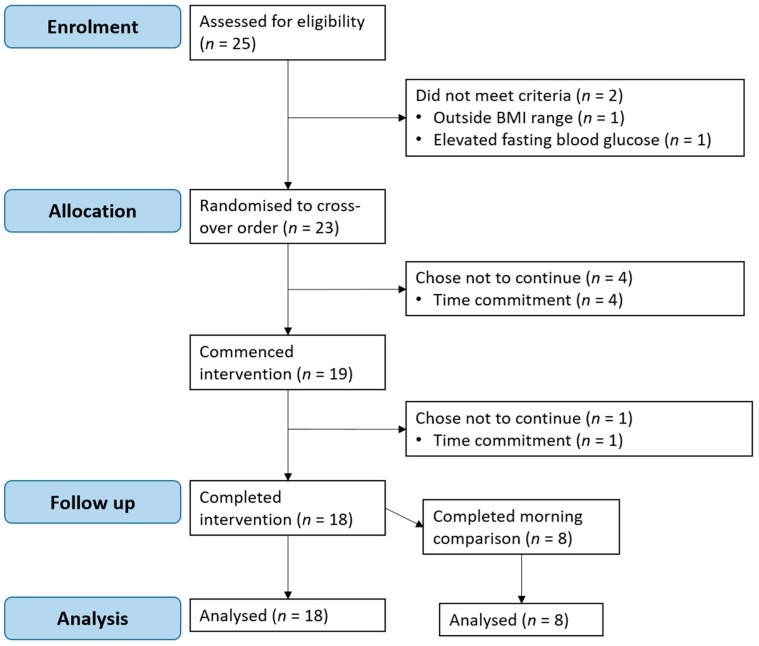
Participant flow diagram for evening postprandial study.

**Figure 3 antioxidants-08-00049-f003:**
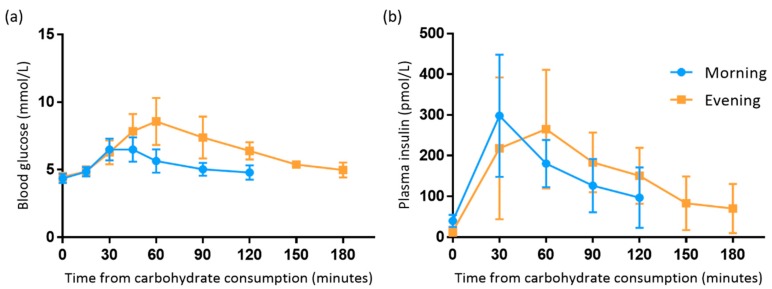
Postprandial blood glucose (**a**) and plasma insulin (**b**) responses following the cellulose treatment in the morning and evening. Data are mean with standard deviation error bars. Paired samples *t* tests showed significant differences between blood glucose iAUC and peak concentration between morning and evening (*p* < 0.05).

**Figure 4 antioxidants-08-00049-f004:**
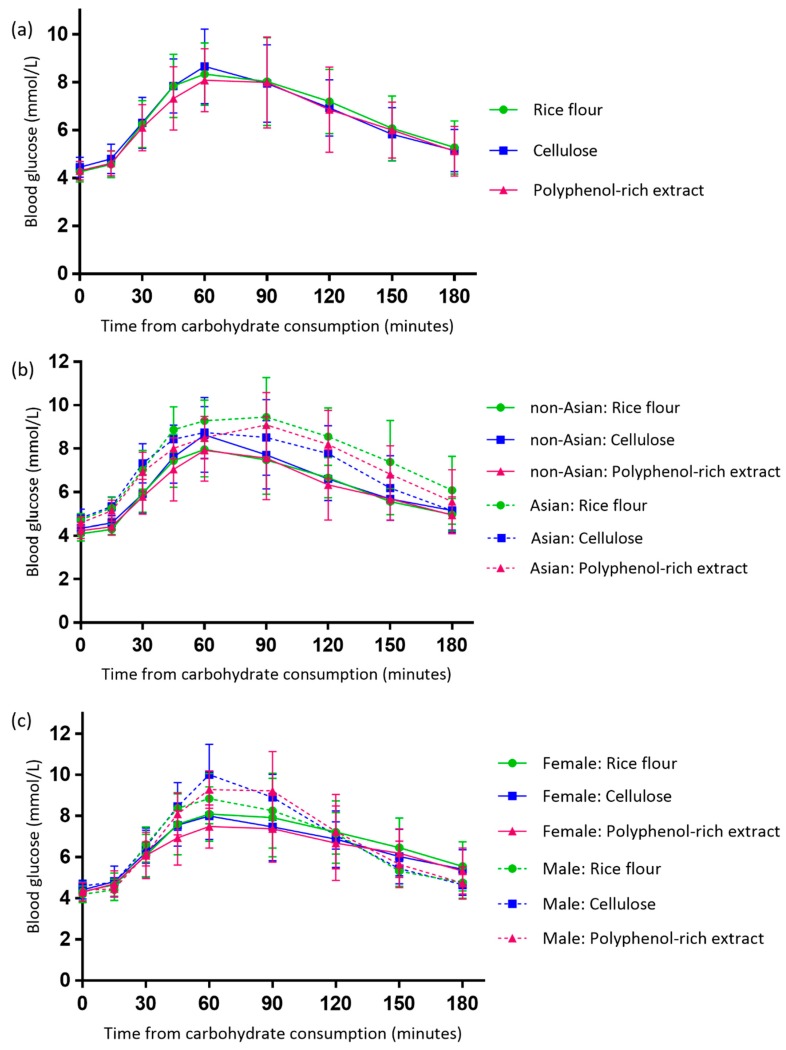
Evening postprandial blood glucose responses following the polyphenol-rich extract, rice flour and cellulose for the total study population (**a**) and grouped by ethnicity (**b**) and sex (**c**). Data are mean with standard deviation error bars. Independent samples *t* tests showed significantly higher peak blood glucose concentrations in males than females following cellulose (*p* = 0.024) and polyphenol-rich extract (*p* = 0.015). A one-way repeated measures ANOVA (analysis of variance) showed significant differences between the three treatments among females (*p* = 0.018).

**Figure 5 antioxidants-08-00049-f005:**
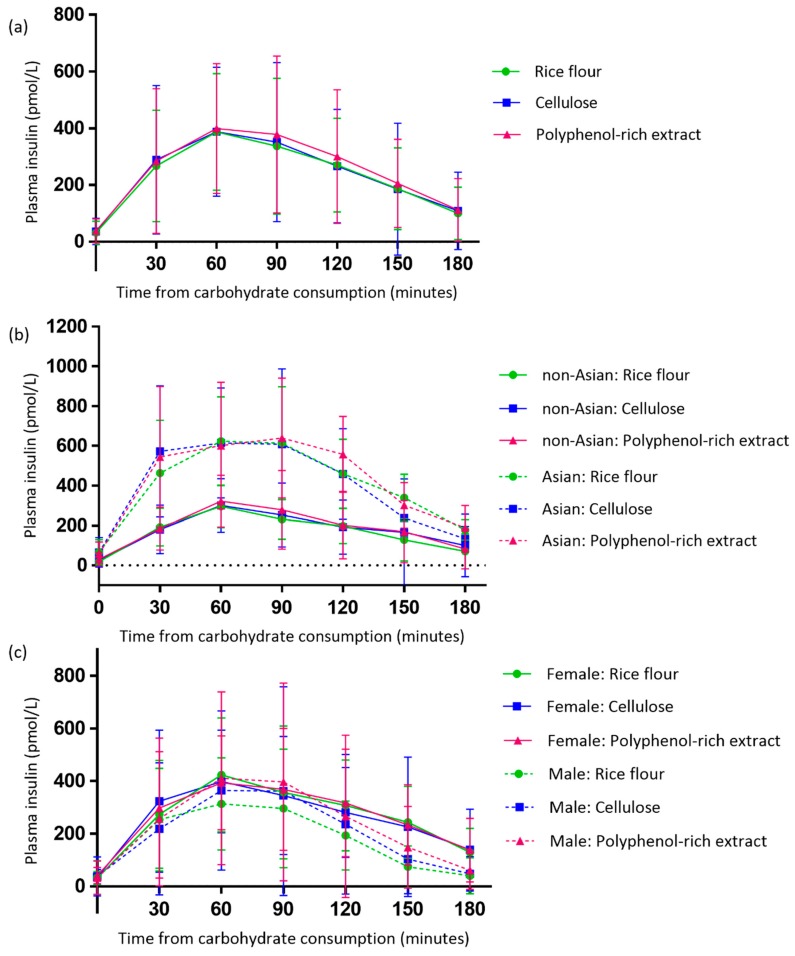
Evening postprandial plasma insulin responses following the polyphenol-rich extract, rice flour and cellulose for the total study population (**a**) and grouped by ethnicity (**b**) and sex (**c**). Independent samples Mann-Whitney U test showed significant differences between Asian and Caucasian participants for iAUC, late phase iAUC and peak concentration following all treatments (*p* < 0.05).

**Table 1 antioxidants-08-00049-t001:** Participant characteristics (*n* = 18).

Characteristic	Mean (SD)
Age (years)	25.5 (19) *
BMI (kg/m^2^)	23.8 (2.6)
Waist circumference (cm)	76.3 (4.4)
Fat mass (%)—females	32.2 (5.8)
Fat mass (%)—males	15.0 (5.3)
Estimated daily polyphenol intake (mg)	1559 (1676) *
Estimated daily energy intake (kJ)	8978 (5508) *
Systolic blood pressure (mmHg)	121 (9)
Diastolic blood pressure (mmHg)	77 (5)
	***n*** **(frequency)**
Female	12 (67%)
Male	6 (33%)
Ethnicity	
Caucasian background	13 (72%)
Asian background	5 (28%)
Physical Activity level	
Moderate	9 (50%)
High	9 (50%)

* Median (IQR); SD, standard deviation; IQR, interquartile range; BMI, body mass index.

**Table 2 antioxidants-08-00049-t002:** Morning and evening comparison subset participant characteristics (*n* = 8).

Characteristic	Mean (SD)
Age (years)	24 (20) *****
BMI (kg/m^2^)	22.5 (2.4)
Waist circumference (cm)	74.9 (5.4)
Percent fat mass—males (%)	12.5 (1.9)
Percent fat mass—females (%)	28.8 (3.0)
Systolic blood pressure (mmHg)	123.8 (8.4)
Diastolic blood pressure (mmHg)	75.8 (4.3)
Male:Female (*n*)	4:4
Caucasian:Asian (*n*)	8:0

***** Median (IQR); SD, standard deviation.

**Table 3 antioxidants-08-00049-t003:** Fasting and postprandial blood glucose measures, for rice flour, cellulose and polyphenol-rich extract in the evening.

Outcome	Group	*n*	Rice Flour	Cellulose	Polyphenol-Rich Extract	*p*-Value ^1^	*p*-Value ^2^
Blood glucose for whole group							
Fasting (mmol/L)	Total	18	4.3 (0.4)	4.5 (0.4)	4.3 (0.4)	0.077	NA
iAUC (mmol/L·3 h)	Total	18	411 (152)	410 (127)	414 (169)	0.881	NA
Peak concentration (mmol/L)	Total	18	8.7 (1.5)	8.9 (1.5)	8.5 (1.6)	0.459	NA
Blood glucose by ethnicity							
Fasting (mmol/L)	Caucasian	13	4.1 (0.3)	4.3 (0.3)	4.2 (0.4)	0.106	**0.007**
	Asian	5	4.7 (0.3)	4.8 (0.4)	4.6 (0.3)	0.084	
iAUC (mmol/L·3 h)	Caucasian	13	368 (130)	399 (140)	376 (177)	0.607	0.118
	Asian	5	521 (162)	439 (93)	515 (105)	0.408	
Peak concentration (mmol/L)	Caucasian	13	8.3 (1.3)	8.8 (1.7)	8.2 (1.7)	0.052	0.177
	Asian	5	9.7 (1.6)	9.3 (1.0)	9.3 (1.2)	0.820	
Blood glucose by sex							
Fasting (mmol/L)	Female	12	4.3 (0.4)	4.4 (0.5)	4.3 (0.5)	0.564	0.800
	Male	6	4.2 (0.4)	4.6 (0.3)	4.4 (0.1)	0.038	
iAUC (mmol/L·3 h)	Female	12	410 (160)	394 (127)	385 (168)	0.529	0.506
	Male	6	412 (149)	443 (132)	473 (170)	0.556	
Peak concentration (mmol/L)	Female	12	8.4 (1.6)	8.4 (1.2) ^a^	7.9 (1.4) ^a^	**0.018**	**0.040**
	Male	6	9.2 (1.1)	10.0 (1.5) ^a^	9.8 (1.3) ^a^	0.168	

All values reported as mean (standard deviation); iAUC incremental area under the curve ^1^ difference between treatment; ^2^ difference between groups; ^a^ Difference between sexes with *p* < 0.05; **Bold** typeface where *p* < 0.05. NA, not applicable.

**Table 4 antioxidants-08-00049-t004:** Fasting and postprandial plasma insulin measures, for rice flour, cellulose and polyphenol-rich extract in the evening.

Outcome	Group	*n*	Rice Flour	Cellulose	Polyphenol-Rich Extract	*p*-Value ^1^
Plasma insulin for whole group						
Fasting (pmol/L)	Total	18	15 (33)	15 (39)	20 (59)	0.978
iAUC (pmol/L·3 h)	Total	18	32,891 (21,899)	27,709 (37,172)	30,085 (47,622)	0.801
Late phase iAUC (pmol/L·90 min)	Total	18	20,483 (17,574)	17,868 (22,933)	20,008 (34,148)	0.846
Peak concentration (pmol/L)	Total	18	331 (299)	318 (351)	348 (401)	0.411
Plasma insulin by ethnicity						
Fasting (pmol/L)	Caucasian	13	7 (22)	7 (32)	7 (60)	0.661
	Asian	5	44 (108)	41 (118)	43 (83)	0.627
*p*-value ^2^			0.059	0.143	0.336	
iAUC (pmol/L·3 h)	Caucasian	13	27,621 (16,297)	23,526 (16,158)	24,526 (26,432)	0.926
	Asian	5	59,288 (46,230)	69,746 (52,200)	70,866 (44,792)	0.549
*p*-value ^2^			**0.003**	**0.019**	**0.007**	
Late phase iAUC (pmol/L·90 min)	Caucasian	13	19,087 (8339)	16,405 (7799)	17,745 (14,203)	0.794
	Asian	5	40,609 (37,736)	50,127 (41,054)	51,391 (30,825)	1.000
*p*-value ^2^			**0.007**	**0.035**	**0.019**	
Peak concentration (pmol/L)	Caucasian	13	283 (114)	280 (210)	317 (222)	0.500
	Asian	5	714 (377)	615 (623)	708 (521)	0.819
*p*-value ^2^			**0.002**	**0.046**	**0.007**	
Plasma insulin by sex						
Fasting (pmol/L)	Female	12	27 (36)	29 (49)	42 (61)	0.558
	Male	6	7 (40)	7 (49)	7 (39)	0.135
*p*-value ^2^			0.291	0.291	0.213	
iAUC (pmol/L·3 h)	Female	12	35,679 (31,396)	30,795 (41,403)	30,085 (47,116)	0.920
	Male	6	23,880 (23,073)	20,070 (39,441)	30,935 (44,374)	0.607
*p*-value ^2^			0.151	0.151	0.616	
Late phase iAUC (pmol/L·90 min)	Female	12	20,811 (16,029)	20,181 (28,424)	20,008 (33,292)	0.920
	Male	6	16,967 (19,008)	14,970 (31,963)	22,825 (34,836)	0.513
*p*-value ^2^			0.250	0.291	0.616	
Peak concentration (pmol/L)	Female	12	333 (327)	377 (339)	348 (406)	0.717
	Male	6	257 (311)	219 (481)	379 (553)	0.513
*p*-value ^2^			0.213	0.151	0.616	

All data are non-parametric reported as median (interquartile range). Late phase insulin response taken as insulin iAUC from 30–120 min postprandial; ^1^ differences between treatments (Friedman’s Two-Way Analysis of Variance by Ranks test); ^2^ differences between ethnic groups/sexes (Independent samples Mann-Whitney U test); **Bold** typeface where *p* < 0.05.

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
