# Peer review of "A Single-Dose of a Polyphenol-Rich Fucus Vesiculosus Extract is Insufficient to Blunt the Elevated Postprandial Blood Glucose Responses Exhibited by Healthy Adults in the Evening: A Randomised Crossover Trial"

_antioxidants, 2019, doi:10.3390/antiox8020049_

Round 1

Reviewer 1 Report

Carbohydrate consumption may elevate and prolong postprandial blood glucose. The authors investigated whether Fucus vesiculosus, a polyphenol-rich algae, can moderate postprandial glycemia. Participants consumed the polyphenol-rich extract prior to a supplement containing 50g carbohydrate at two different time points (morning or evening). There was no difference in peak postprandial ghycemia at night compared to placebo overall, however the authors report that the peak blood glucose concentration was reduced in females. These results may warrant further investigation of the polyphenol-rich algae on lowering postprandial hyperglycemia. This is an interesting study that had potentially identified possible differences in gender and ethnicity which may warrant further investigation. While interesting, the manuscript requires editing for grammar and spelling mistakes.

Author Response

Thank you for your positive feedback, the manuscript has been checked for grammar and spelling.

Reviewer 2 Report

This study attempted to show an effect of polyphenols in the extract from Fucus vesiculosus on elevation of postprandial blood glucose level. However, it was not unfortunately shown the warrants in this article. The authors have considered that this study showed an efficacy of the randomised crossover trial. However, the reviewer considers it important that reducing effect of the polyphenol on blood glucose level will be shown. Or an inhibition effect on weight increase could be fine, too, in the body which develops diabetes.

Author Response

Thank you for your feedback.  However, the finding of the study was that the polyphenol-rich extract had no effect on postprandial glucose, and it is good science to report this.  A lowering effect was observed on peak postprandial glucose concentration among females and this is commented on in the paper.

In regards to your comment about effect on weight.  An effect on weight can only be shown in longer-term interventions, whereas this paper reported an acute postprandial trial, so weight was not an outcome of interest. 

Reviewer 3 Report

Antioxidants-425343-v1

Murray et al. investigated the “A single-dose of a polyphenol-rich Fucus vesiculosus extract is insufficient to reduce the elevated postprandial blood glucose responses exhibited by healthy adults in the evening: a randomised crossover trial”.

Several comments for this manuscript:

1.      First page “article title” incorrect format.

2.      The authors can further re-organize the abstract to be more concise and conclusive, it is not so clear in this version.

3.      In the text : Marine algae contain unique bioactive polyphenolic molecules with capacity to moderate postprandial hyperglycaemia. Please indicate what the ingredients belong to polyphenols? in this study.

4.      In results, lack of relevant mechanisms to prove how the vesiculosus extract really lower postprandial blood glucose responses? Whether through impact phosphorylation of AMPK, ACC and Akt or glucose uptake. Although Vesiculosus extract is insufficient to reduce the elevated postprandial blood glucose responses exhibited by healthy adults.

5.      Keywords and the topic is not appropriate. For example, type 2 diabetes. However, this study focus on healthy adults.

6.      References section, please make the format of the references consistently and follow the author’s instructions of this journal.

Author Response

1.                   First page “article title” incorrect format.

The article title has been adjusted to the required format and now reads as follows:

“A Single-Dose of a Polyphenol-Rich Fucus Vesiculosus Extract is Insufficient to Blunt the Elevated Postprandial Blood Glucose Responses Exhibited by Healthy Adults in the Evening: a Randomised Crossover Trial”

2.                   The authors can further re-organize the abstract to be more concise and conclusive; it is not so clear in this version.

Thank you for your feedback, the abstract has been re-organised and now reads as follows:

“When healthy adults consume carbohydrates at night, postprandial blood glucose responses are elevated and prolonged compared to daytime, and extended postprandial hyperglycaemia is a risk-factor for type 2 diabetes. Polyphenols are bioactive secondary metabolites of plants and algae with potential to moderate postprandial glycaemia.  This study investigated whether a polyphenol-rich algae (Fucus vesiculosus) extract moderated postprandial glycaemia in the evening in healthy adults.  In a double blind, placebo-controlled, randomised three-way crossover trial, 18 participants consumed a polyphenol-rich extract, a cellulose placebo and rice flour placebo (7:15 pm) prior to 50 g available carbohydrate from bread (7:45 pm), followed by three hours of blood sampling to assess glucose and insulin.  A subset of participants (n = 8) completed the same protocol once in the morning with only the cellulose placebo (7:15 am).  No effect of the polyphenol-rich extract was observed on postprandial glycaemia in the evening, compared with placebos, in the group as a whole, however in females only, peak blood glucose concentration was reduced following the polyphenol-rich extract.  In the subset analysis, as expected, participants exhibited elevated postprandial blood glucose in the evening compared with the morning following the cellulose placebo.  This was the first study to investigate whether a polyphenol intervention moderated evening postprandial hyperglycaemia.  The lowering effect observed in females suggests that this warrants further investigation.”

3.                   In the text : Marine algae contain unique bioactive polyphenolic molecules with capacity to moderate postprandial hyperglycaemia. Please indicate what the ingredients belong to polyphenols? in this study.

Thank you for this suggestion, the following detail has been added to the introduction:

“Marine algae contain bioactive polyphenolic molecules with capacity to moderate postprandial hyperglycaemia, including phlorotannins, which are unique to marine algae[4].”

Unfortunately, we did not have capacity to characterise the specific polyphenolic molecules within the extract.  Therefore, we can only comment on the effects of the extract as a whole, rather than specific polyphenols.  The best we can do is state that the extract included molecules from the class of polyphenols; phlorotannins.

4.                   In results, lack of relevant mechanisms to prove how the vesiculosus extract really lower postprandial blood glucose responses? Whether through impact phosphorylation of AMPK, ACC and Akt or glucose uptake. Although Vesiculosus extract is insufficient to reduce the elevated postprandial blood glucose responses exhibited by healthy adults.

Thank you for your comment, however we disagree that the mechanisms should be commented on in the results section.  The mechanisms of action were not a finding of this paper so it is not correct to include details of this in the results.  Not did we think it was appropriate to focus on mechanisms in the discussion, as the study showed largely negative results.  Instead, we focused on potential limitations and possible reasons for the findings in the discussion.

Our hypothesis was based on research from in vitro, cell and animal models that demonstrated potential anti-hyperglycaemic mechanisms.  The discussion of this in the introduction has been strengthened as follows:

“Marine algae contain bioactive polyphenolic molecules with capacity to moderate postprandial hyperglycaemia, including phlorotannins, which are unique to marine algae[4].  Potential mechanisms of action have previously been reviewed[4] and include the inhibition of carbohydrate digestive enzymes, α-amylase and α-glucosidase, as demonstrated in chemical assay[4], and the alteration of hepatic enzyme activity (inhibiting glucose-6-phosphatase and phosphoenolpyruvate carboxykinase) to promote glycogen production and the removal of glucose from the blood, as demonstrated in a diabetic mouse model[7].  Marine algal polyphenols (MAPs) have also been shown to upregulate phosphorylation of AMPK (adenosine monophosphate-activated protein kinase), ACC (acetyl-CoA carboxylase) and Akt in diabetic mouse and rat models to increase the number of GLUT4 (glucose transporter 4) transporters at the cell membrane and increase glucose uptake at a cellular level[8, 9].”

5.                   Keywords and the topic is not appropriate. For example, type 2 diabetes. However, this study focus on healthy adults.

This is a relevant point.  The keyword ‘type 2 diabetes’ has been removed replaced with ‘hyperglycaemia’ and ‘hyperinsulinaemia’.

6.                   References section, please make the format of the references consistently and follow the author’s instructions of this journal.

Thank you, we have checked that the references are consistent and ensured that the style matches the journal instructions.

Reviewer 4 Report

The strengths and limitations of the study should be deeply addressed, taking into account sources of potential bias or imprecision: Discuss both direction and magnitude of any potential bias.

The number of studied subjects must be expanded.

The differences in outcome between males and females should be better addressed.

Author Response

The strengths and limitations of the study should be deeply addressed, taking into account sources of potential bias or imprecision: Discuss both direction and magnitude of any potential bias.

Thank you for your feedback.  The strengths and limitations sections has been updated, as shown below, to more deeply discuss the effect of any limitations on the results.  Bias is unlikely due to the rigorous study design used (as described in the manuscript), including double-blinding, use of two placebos, randomisation and a crossover design.  Any potential imprecision in the results would be due to the equipment, however all practicable steps were taken to improve precision: glucose was measured using a calibrated HemoCue and insulin was assessed in duplicate with the coefficient of variation reported in the manuscript (“Across all plates, the mean coefficient of variation was 8.5% (standard deviation (SD) 15.3).”).  We also used a finger prick methodology (rather than venous blood sampling) for assessing postprandial glucose, which is the gold standard.

The strengths and limitations section now reads as follows:

“This was the first study to investigate the efficacy of a polyphenol-based intervention at moderating the postprandial hyperglycaemia that occurs in the evening. A strength of this trial was the randomised, double-blind, placebo-controlled, crossover nature of the study design.  Furthermore, the use of both a cellulose fibre and non-fibre (rice flour) placebo accounted for the fibre (fucoidan) content of the extract to show the effect of the polyphenol content of the extract alone, and provided a comparator to assess the effect of the extract as a whole.  Another strength of this research was the investigation of postprandial responses based on ethnicity and sex.  These findings indicated clear differences in insulin metabolism between individuals of Asian and Caucasian backgrounds, and differences in peak postprandial glucose concentrations between males and females, even in a relatively small sample.  In addition, the comparison between postprandial responses in the evening and the morning supports and builds on evidence for the effects of meal timing on metabolic outcomes[17, 18, 22, 26].   A potential limitation of the study was the relatively small sample size (18 participants) which puts a limit on the applicability of the observations regarding ethnicity and sex until research with larger samples confirms these findings.  Another potential limitation was the use of healthy participants, which may have diminished the glycaemic lowering potential of the polyphenol-rich extract due to individuals already having adequate glucose regulation.  Further research in populations with pre-diabetes or T2DM may observe effects of greater magnitude.  This research provides pilot data that future studies investigating polyphenol-based interventions on evening glycaemic responses can use to calculate power.”

The number of studied subjects must be expanded.

The number of included subjects was determined based on a power calculation.  While no effect was observed with this number of subjects in this study, this may serve as data for future studies to base their power calculations on.

The differences in outcome between males and females should be better addressed.

Thank you for your comment.  Given that the study had a relatively small sample size, we did not want to over emphasise the significance of the differences observed between men and women.  We have included some more discussion of the differences between males and females in peak postprandial glucose concentration, as shown below, while still not over emphasising these findings.

“Another sex difference observed in the present study was that male participants had significantly higher peak postprandial blood glucose levels compared with female participants.  This is in contrast to evidence which suggests that females are more likely to experience elevated blood glucose concentrations following an OGTT[40,46,47], as a consequence of generally having a smaller body size and thus less muscle mass, but being given the same glucose load[47].  Likewise, in the present study, females had a smaller body size than males (on average 13.7 kg less fat free mass than males), which therefore does not explain the differences observed.  It is also unlikely to be a consequence of age or visceral fat as there were no significant differences between males and females for age, BMI or visceral fat volume.  This finding may be due to differences between the sexes in the way that glucose homeostasis is maintained[40]. Males typically have lower insulin sensitivity than females, which is countered by their greater muscle mass and lower body fat mass[40].  Reduced insulin sensitivity results in less efficient uptake of glucose by the tissues and less efficient inhibition of endogenous glucose production which may explain the elevated postprandial blood glucose responses observed in males[48].”

Round 2

Reviewer 2 Report

The reviewer understands the authors’ idea which is no effect is a science, too, and agrees with it. Nevertheless, the authors should show the availability of the extract as antioxidant according to the aim and scope of this journal, or antioxidative properties of the alga extract. The reviewer did not think that only decreasing glucose level on peak in female was sufficient.

Author Response

Thank you for this suggestion.  Along with discussion of the effects of the extract on postprandial glucose and insulin, we have added to the manuscript a description of a Folin-Ciocalteu assay analysis of the extract to quantify the total polyphenol (antioxidant) content of the extract, as below:

Methods

2.7 Quantification of soluble polyphenols

An adapted Folin-Ciocalteu methodology was used to quantify the total soluble polyphenols in the extract [37] with phloroglucinol dihydrate used as standard (Sigma-Aldrich P38005). The extract was dissolved in 10 mL of distilled water and diluted to reach concentrations of 25, 50 and 100 µg/mL. The assay was performed by pipetting 2 mL of distilled water (blank), the phloroglucinol standards (5, 10, 15, 20, 30, 50 and 100 µg/mL), and the sample solutions into sequential vials.  Folin-Ciocalteu reagent (500 µL) (Sigma-Aldrich F9252) was then added to each vial and allowed to stand for 5 minutes.  Then 1500 µL of 7.5% w/w sodium carbonate solution and 4000 µL of distilled water were added to each vial.  The reaction was then incubated in the dark at room temperature for two hours.  Analysis was conducted using a spectrophotometer at 765 nm, with the solutions in quartz cuvettes.  All samples, standards and blanks were run in triplicate and absorbance values were recorded.

Results

3.6 Polyphenol content of extract

The total soluble polyphenol concentration of the extract was determined to be 29.7%, according to the Folin-Ciocalteu assessment of water extracts of powdered Fucus vesiculosus.  This concentration is comparable to the suggested 28% polyphenol content in the extract, as certified by the supplier.  The mean coefficient of variation among the triplicates was 2.5%.  

Reviewer 3 Report

The advice have been corrected.

Author Response

Thank you.

Round 3

Reviewer 2 Report

The reviewer appreciates the response and effort which has been done by the authors to enhance the quality of the work. After the check and correction of misspelling is done, the manuscript would be acceptable.